# Lateral Involvement in Different Sized Papillary Thyroid Carcinomas Patients with Central Lymph Node Metastasis: A Multi-Center Analysis

**DOI:** 10.3390/jcm11174975

**Published:** 2022-08-24

**Authors:** Yu Heng, Zheyu Yang, Pengyu Cao, Xi Cheng, Lei Tao

**Affiliations:** 1ENT Institute and Department of Otorhinolaryngology, Eye & ENT Hospital, Fudan University, Shanghai 200031, China; 2Department of General Surgery, Ruijin Hospital, Shanghai Jiaotong University School of Medicine, Shanghai 200031, China

**Keywords:** papillary thyroid microcarcinoma, central lymph node metastasis, lateral lymph node metastasis, maximum tumor diameter, risk stratification

## Abstract

Objective: To quantitatively predict the probability of lateral lymph node metastasis (LLNM) for papillary thyroid carcinomas (PTC) patients with central lymph node metastasis (CLNM) in order to guide postoperative adjuvant treatment. Methods: Five hundred and three PTC patients with CLNM from three medical centers were retrospectively analyzed. Results: The LLNM rate for all patients was 23.9% (120 in 503), with 15.5% (45 in 291) and 35.4% (75 in 212) for patients with papillary thyroid microcarcinoma (PTMC) and large papillary thyroid carcinoma (LPTC), respectively. Patients with no fewer than five positive central lymph nodes (CLN) exhibited a higher risk of LLNM. For patients with fewer than five positive CLN, a maximum diameter of positive CLN > 0.5 cm and the presence of ipsilateral nodular goiter were identified as independent risk factors of LLNM for papillary thyroid microcarcinoma (PTMC) patients. The independent risk factors of LLNM for large papillary thyroid carcinoma (LPTC) patients included a tumor located in the upper portion of thyroid, maximum tumor diameter ≥ 2.0 cm, maximum diameter of positive CLN > 0.5 cm, and the presence of thyroid capsular invasion. Predictive nomograms were established based on these risk factors for PTMC and LPTC patients, respectively. The accuracy and validity of our newly built models were verified by C-index and calibration curves. PTMC and LPTC patients with fewer than five positive CLN were each stratified into three subgroups based on their nomogram risk scores, and a detailed risk stratification flow chart was established for a more accurate evaluation of LLNM risk in PTC patients. Conclusions: A detailed stratification flow chart for PTC patients with CLNM to quantitatively assess LLNM risk was established, which may aid in clinical decision-making for those patients.

## 1. Introduction

As the most common malignancy of the endocrine system, the incidence of thyroid carcinoma has shown a constant increase in recent years [1,2]. Papillary thyroid carcinoma (PTC) is the most prevalent among all pathological types of thyroid cancer. Due to its relatively indolent characteristic, patients with PTC often exhibit satisfactory prognosis in long-term follow-up [3]. However, cervical lymph node metastasis (LNM) occurs frequently in PTC, with reported occurrence rates ranging from 40–90% [4]. Even patients in the early stages of PTC can suffer from LNM [5]. LNM is also regarded as an independent risk factor for tumor recurrence after initial surgery for PTC patients [6,7,8], making it an important aspect of cervical management.

Usually, the cervical lymph node metastases of PTC patients occur first in the central compartment and subsequently in the lateral neck [8]. For patients with preoperatively detected CLNM by ultrasonography or fine-needle aspiration (FNA) examinations, surgical intervention covering both primary tumor site and central lymph node regions would be performed to avoid postoperative recurrence. This is the approach worldwide. In addition, prophylactic central lymph node dissection (CLND) is also recommended by the Japanese Society of Thyroid Surgeons for patients with negative central lymph node metastasis (CLNM) based on the probability of occult CLNM, with incidence rates ranging from 30–80% [9,10]. Some researchers also suggest that prophylactic CLND should be routinely conducted for cases with T3 or T4 tumors [11]. For the management of lateral cervical compartment, studies have revealed that 18.6% to 39.5% of PTC patients may present with occult lateral lymph node metastasis (LLNM) [12,13], and up to 21.1% of papillary thyroid microcarcinoma (PTMC) patients could have LLNM [14]. Conventionally, PTMC is perceived as a low-risk subgroup in terms of lateral neck involvement. On the other hand, the diagnostic sensitivity of preoperative ultrasonography for LLNM was reported to be lower than 30% [15], indicating the difficulty of LLNM detection. Therefore, some LLNM in PTC patients may be ignored and present as cervical lymph node recurrence during postoperative follow-up. However, considering the increased risk of postoperative complications resulting from lateral lymph node dissection (LLND), prophylactic LLND is generally not accepted as a standard strategy for patients with PTC [16]. Reflecting on these considerations, it would be unwise to either blindly oppose or routinely extend the usage of prophylactic LLND, warranting further meticulous risk assessment of lateral neck involvement for PTC patients with positive CLNM to guide individualized surgical strategies. This may be valuable in avoiding unnecessary surgery-related complications that may result from prophylactic LLND.

Numerous studies have focused on overall LNM or CLNM [9,17]. Yet, studies targeting LLNM in PTC patients are limited, only qualitatively summarizing the risk factors of LLNM in patients with general PTC [5,6,15] and lacked more refined stratifications based on the patients’ characteristics. Here in our study, a meticulous evaluating system that can efficaciously quantify risks of LLNM for PTC with different clinical features was established.

## 2. Materials and Methods

### 2.1. Patient Cohort

Between 2018 and 2020, 568 thyroid carcinoma patients received initial surgery at the following three hospitals: Department of Otorhinolaryngology, Head and Neck Surgery at the Eye, Ear, Nose and Throat Hospital of Fudan University; the Department of General Surgery at Ruijin Hospital of Shanghai Jiao Tong University School of Medicine; and the Department of General Surgery, Civil Aviation Shanghai Hospital. All patients were diagnosed as positive CLNM by postoperative pathological analyses. Patients with any of the following criteria were excluded from this research: (1) non-PTC pathological type (*n* = 41); (2) having received thyroid-related surgery previously (*n* = 16); and (3) history or coexistence of other primary tumors (*n* = 8). Following these criteria, a total of 503 patients were included and analyzed. This study was approved by the Institutional Ethics Committee of the Eye & ENT Hospital of Fudan University and the Ruijin Hospital of Shanghai Jiao Tong University School of Medicine.

### 2.2. Surgical Management

Clinicopathological data were collected to set up the retrospective database. A total of 280 (55.7%) patients received thyroid lobectomy and the other 223 (44.3%) patients received total thyroidectomy. Central lymph node dissection was conducted for all patients enrolled. Therapeutic LLND was performed for those with clinically detected LLNM using both preoperative ultrasonography and fine-needle aspiration (FNA). The prophylactic LLND was conducted only for those with clinical detected lateral lymph nodes that were highly suspected as having tumor involvement using preoperative ultrasonography but later proven LLNM negative by FNA biopsy. Patients enrolled were treated with postoperative TSH suppression therapy and RAI (Radioactive Iodine) therapy according to the 2015 ATA Guidelines. The pathological diagnoses were confirmed by at least two board-certified pathologists. For patients who received CLND only, if positive LLNM were found by ultrasonography and FNA within six months after initial surgery, they will be regarded as having lateral involvement at the time of operation. The thyroid glands were categorized into three equal volumes (upper portion, middle portion, and lower portion) based on the consensus of most clinical medical centers. Tumors with a maximum diameter of more than 2 cm that were primarily located in the upper portion and did not exceed the lower 1/3 thyroid gland were also defined as upper portion tumor in this study.

### 2.3. Statistical Analyses

Chi-square and independent t-tests were used for comparing categorical and continuous variables, respectively. Logistic univariate and multivariate regression analyses were used to select risk factors that were significantly correlated with LLNM by the SPSS 24.0 package (SPSS Inc., Chicago, IL, USA). Variables screened by multivariate analysis were further used for the establishment of a risk prediction model nomogram, performed by R software (version 3.5.1; R Development Core Team, Vienna, Australia). Then the discrimination and consensus degree of our newly established predictive model was tested by the concordance index (C-index), receiver operating characteristic (ROC) curve, and the calibration curve.

## 3. Results

### 3.1. Characteristics, Surgical Procedure, and LLNM of Patients in Our Cohort

Five hundred and three PTC patients with pathologically confirmed CLNM were enrolled in this research. Those with tumors measuring 1.0 cm or less were classified as papillary thyroid microcarcinoma (PTMC), and others with tumors measuring larger than 1.0 cm were defined as large papillary thyroid carcinoma (LPTC). As a result, 291 (57.9%) patients were confirmed to have PTMC and 212 (42.1%) patients were LPTC. All patients received a total thyroidectomy or thyroid lobectomy. CLND was conducted routinely for all patients in our study regardless of whether metastases were detected before operation. However, LLND was performed only for those with positive or highly suspicious LLNM by surgeons during intraoperative phases. As a result, 122 (24.3%) patients received LLND, with 46 (15.8%) in the PTMC group and 76 (35.8%) in the LPTC group. Of the 46 patients that received LLND in the PTMC group, 32 had preoperative confirmed LLNM, and 14 of them were considered high-risk LLNM during operation with 6 confirmed by postoperative pathology. Seven of the 245 PTMC patients receiving CLND alone were detected to have LLNM within six months in post-operation follow-up. In total, 45 (15.5%) patients in the PTMC group were regarded as having LLNM before surgery in our cohort. For the LPTC group, 56 of the 76 patients that received LLND had preoperative confirmed LLNM, and for the other twenty patients that received LLND for prophylactic purposes, nine were confirmed to have LLNM according to postoperative pathology. Ten of the one hundred and thirty-six LPTC patients receiving CLND alone were detected to have LLNM within six months after the initial surgery. In total, 75 (35.4%) patients in the LPTC group were considered as having preoperative LLNM (as shown in Figure 1).

The clinicopathological characteristics of patients enrolled are shown in Table 1. Thyroid capsular invasion (TCI), number of positive central lymph node (CLN) ≥ 5, and Maximum diameter of positive CLN ≥ 1.0 cm were significantly more common in the LPTC group (*p*-value < 0.05). Moreover, the incidence rate of LLNM in the LPTC group was 35.4% (75 in 212), which was significantly higher than that of patients in the PTMC group (15.5%, 45 in 291, *p*-value < 0.05).

### 3.2. Comparison between Patients with LLNM or Not for Patients within Different Groups

Comparisons between patients with CLNM alone and those with LLNM were made for PTMC and LPTC patients, respectively (shown in Table 2). The results showed that the multifocality and bilateral disease of the thyroid were significantly more frequently detected in patients with positive lateral neck involvement than in those with CLNM alone in patients within the PTMC group (*p*-value = 0.000 and 0.030, respectively), while no difference was found between patients with LLNM or without LPTC. On the contrary, thyroid capsular invasion and the upper portion tumor of thyroid (*p*-value = 0.000 and 0.030, respectively) were significantly more common in patients with LLNM than in those with CLNM alone for LPTC rather than PTMC patients. Meanwhile, for patients within the LPTC group, those with LLNM showed significantly bigger tumor size than those with CLNM alone (2.03 ± 1.09 cm vs. 1.58 ± 0.62 cm, *p*-value = 0.000). In addition, the number of positive CLN ≥ 5, the maximum diameter of positive CLN ≥ 1.0 cm, and PTC with ipsilateral nodular goiters (iNG) were more common in LLNM patients for both PTMC and LPTC groups.

### 3.3. Postoperative Complications of Patients Receiving CLND Alone and Patients Receiving CLND + LLND

The surgery-related complications of patients receiving CLND alone and CLND + LLND were analyzed (shown in Table 3). In terms of temporary postoperative complications including a temporary hypoparathyroid hormone and hoarseness, no difference was shown between patients receiving LLND or not (*p*-value = 0.646 and 0.247, respectively). However, patients receiving CLND + LLND showed higher permanent hypoparathyroid hormone (4.9% vs. 1.8%, *p*-value = 0.062) and hoarseness (4.1% vs. 1.6%, *p*-value = 0.097) rates than those receiving CLND alone, although the differences were not statistically significant. The incidence rate of chyle leakage was significantly higher in patients receiving CLND + LLND than in patients with CLND alone (0.2% vs. 3.3%, *p*-value = 0.003). In general, only 3.7% (14 in 381) of patients receiving CLND alone developed relatively severe surgery-related complications. These include a permanent hypoparathyroid hormone, permanent hoarseness, and postoperative chyle leakage. For patients that received CLND + LLND, 12.3% (15 in 122) developed these symptoms, which was significantly higher (*p*-value = 0.000).

### 3.4. Patients with No Fewer Than Five Positive Central Lymph Nodes Exhibited a Significantly Higher LLNM Rate Than the Others

It has been reported that PTC patients with more positive central lymph nodes were more likely to develop LLNM [18,19]. We found that patients with a number of positive CLN no fewer than five exhibited a high probability of LLNM, 35.2% (19 in 54) in the PTMC group and 52.6% (40 in 76) in LPTC patients, both significantly higher than those with fewer than five positive CLN (11.0% (26 in 237) for PTMC group and 25.7% (35 in 136) for LPTC group, *p* = 0.000 and 0.000, respectively). Those with no fewer than five positive CLN were thus categorized as high-risk subgroup for LLNM in PTC patients. However, for patients who exhibited fewer than five positive CLN, PTMC and LPTC groups showed significantly different incidence rates in terms of LLNM (11.0% (26 in 237) and 25.7% (35 in 136), respectively, *p*-value = 0.000).

### 3.5. Risk Stratification and Validation for Lateral Involvement in PTMC Patients with Fewer Than Five Positive Central Lymph Nodes

Univariate and multivariate logistic regression analyses were then performed to screen out independent risk factors of LLNM for PTMC patients with fewer than five positive CLN. The factors including more than one positive CLN, maximum diameter of positive CLN > 0.5 cm, and the presence of ipsilateral nodular goiter and multifocality were confirmed to be associated with LLNM and were further enrolled in the multivariate analysis (shown in Table 4). Results yielded indicated that maximum diameter of positive CLN > 0.5 cm and the presence of ipsilateral nodular goiter were independent risk factors of LLNM for PTMC patients with fewer than five positive CLN. The two selected factors were then applied to construct the prediction nomogram model of LLNM for these patients (shown in Figure 2A).

For the evaluation and validation of the nomogram, we conducted an internal validation by 1000 bootstrap resamples to assess the prediction accuracy of LLNM for PTMC with fewer than five positive CLN in terms of C-index. The C-index turned out to be 0.912 (95% CI, 0.851–0.972) and 0.906 (95% CI, 0.890–0.922) after bootstrapping, indicating the satisfactory accuracy of our newly-established model. The corresponding ROC curve was shown in Figure 2C. Furthermore, the calibration plot also proved that the actual and estimated probability of LLNM in PTMC patients with fewer than five positive CLN were in fair agreement (Figure 2E).

Both of the two factors that constituted the nomogram had their corresponding risk points. The risk scores of two factors were summed up for the 237 PTMC patients with fewer than five positive CLN. Based on the distribution of the total score, we divided these patients into three subgroups using two cutoff values:(1)a low-risk subgroup (with LLNM risk score of =0, *n* = 135),(2)a moderate-risk subgroup (0 < LLNM risk score of <100, *n* = 49),(3)a relatively high-risk subgroup (with LLNM risk score ≥ 100, *n* = 53).

The LLNM rates of low-, moderate-, and high-risk subgroups were 0.7 % (1 in 135), 6.1% (3 in 49), and 41.5% (22 in 53), respectively, showing significantly different LLNM risks among the three subgroups (*p*-value = 0.000, shown in Table 5).

### 3.6. Risk Stratification and Validation for Lateral Involvement in LPTC Patients with Fewer Than Five Positive Central Lymph Nodes

For patients with LPTC, those exhibiting fewer than five positive CLN were also enrolled for univariate and multivariate analyses (Table 6). Four factors including tumor located in the upper portion of thyroid, maximum tumor diameter ≥ 2.0 cm, maximum diameter of positive CLN > 0.5 cm, and the presence of thyroid capsular invasion were identified as independent risk factors of LLNM for these patients and were used to establish a prediction model (shown in Figure 2B). The C-index yielded 0.826 (95% CI, 0.753–0.900), and 0.817 (95% CI, 0.799–0.835) after bootstrapping (the ROC curve and the calibration plot were shown in Figure 2D,E).

The risk scores of four factors were summed up for the 136 LPTC patients with fewer than five positive CLN, and two cutoff values were also selected for these patients:(1)a low-risk subgroup (LLNM risk score < 100, *n* = 46),(2)a moderate-risk subgroup (100 ≤ LLNM risk score of <150, *n* = 26),(3)a relatively high-risk subgroup (LLNM risk score ≥ 150, *n* = 64).

The LLNM rates of low-, moderate-, and high-risk subgroups were 2.2% (1 in 46), 15.4% (4 in 26), and 46.9% (30 in 64), respectively, also showing a significant difference in LLNM risk among the three subgroups (*p*-value = 0.000, shown in Table 5).

### 3.7. Detailed Risk Stratification Flow Chart for PTC Patients with Positive CLNM

Based on the aforementioned classification for LLNM in patients with different tumor progressions, a detailed LLNM risk stratification flow chart was created for all PTC patients with positive CLNM and is shown in Figure 3.

## 4. Discussion

There is an ongoing debate about the optimal strategy of lateral neck management for PTC patients with positive CLNM. Here in our study, a meticulous risk assessment system was created for these patients to quantitatively assess the lateral neck involvement risk. Patients with ≥5 positive CLN was classified into high-risk group based on their significantly higher rate of LLNM than other patients.

For those with <5 positive CLN, PTMC and LPTC patients with evaluation scores no less than 100 and 150 according to their respective nomogram were also categorized into the high-risk group, both with LLNM rates of over 40%. On the contrary, those with evaluation scores = 0 and less than 100 according to nomograms for PTMC and LPTC patients, respectively, were stratified as a low LLNM risk group due to the extremely low lateral neck involvement rates (0.7% and 2.2% for PTMC and LPTC patients, respectively).

Previous studies have revealed that patients with more than five metastatic lymph nodes showed significantly stronger tendencies towards LLNM and could be used as a predictive index for lateral neck involvement in PTC patients [18], which was consistent with our study. LLNM rates in patients with ≥5 positive CLN was 45.4% for all patients, while it was 35.2% and 52.6% for PTMC and LPTC groups, respectively. Considering that nearly half of the patients exhibited LLNM, those with no fewer than five positive lymph nodes were regarded as LLNM high-risk patients and were separated from all patients to further screen out high-risk patients within those with <5 positive CLN.

Several previous studies have regarded tumor size as an independent risk factor for LLNM [5,15,20,21]. Those with larger tumor sizes also showed significant tendencies towards tumor recurrence compared to those with small tumor sizes [22]; a reminder that careful consideration should be taken in the selection of surgical procedures in the management of PTC patients with different primary tumor sizes. Our study also indicated that for patients who exhibited fewer than five positive CLN, significantly different incidence rates in terms of lateral neck involvement (11.0% vs. 25.7%) was found for the PTMC and LPTC groups. Thus, the two groups will be discussed separately.

PTMC is generally defined as a low degree of invasive tumor considering their quite indolent nature. Recently, an active surveillance strategy is considered an alternative to immediate resection for patients with low-risk PTMC who exhibit no clinical evidence of local metastases [23,24]. Even for those receiving surgical treatment, a “wait and see” strategy is enough and prophylactic LND is unnecessary at most clinical centers. However, few reports investigated PTMC patients with lymph node metastases. The existing literature shows that several factors are significantly associated with LLNM in PTMC patients, including tumors located in the upper portion, the number of CLNM, extrathyroidal extension, and multifocal lesions [25,26,27]. Based on this, we wondered whether there is a unique subgroup of patients with a high risk of lateral neck metastasis in this traditionally considered low-risk population, which would have significant implications for the decision-making of lateral neck management. Here in our research, we used statistical methods to screen out risk factors for PTMC patients with <5 positive CLN. The results showed that the maximum diameter of positive CLN > 0.5 cm and the presence of ipsilateral nodular goiter (iNG) were identified as independent risk factors of LLNM for these patients. A predictive nomogram was then established, and all patients enrolled received a total score that could efficiently quantify their risk of LLNM. Three subgroups were then formed according to the nomogram score for further division of these patients. Those with a total score of 0 were classified as low-risk LLNM subgroup, where only 1 in 135 (0.7%) patients in this subgroup exhibited LLNM. This means those with a maximum diameter of positive CLN ≤ 0.5 cm and negative iNG within PTMC patients exhibiting < 5 positive CLN, the incidence rate of LLNM is extremely low; thus, LND covering the central lymph node region is enough. However, although the majority of the PTMC patients with <5 positive CLN were categorized into low the LLNM risk subgroup (135 in 237, 57.0%), a small portion of patients with nomogram scores ≥ 100 were screened out and were regarded as a high-risk subgroup of LLNM within these patients, with LLNM rate of 41.5% (22 in 53).

An investigation was also conducted on LPTC patients with fewer than five positive CLN in our research. Unlike those of the PTMC group, iNG showed no significant association with LLNM in the LPTC group, while four factors including tumor located in the upper portion of thyroid, maximum tumor diameter (MTD) ≥ 2.0 cm, maximum diameter of positive CLN > 0.5 cm, and the presence of thyroid capsular invasion (TCI) were proven to be independent risk factors of LLNM for patients within LPTC group, which further indicates the important implications of our study to separate patients by different tumor sizes. LPTC patients with fewer than five positive CLN were also divided into low- (total score < 100), moderate- (100 ≤ total score < 150), and high- (total score ≥ 150) risk LLNM subgroups. The incidences of LLNM rates were 2.2%, 15.4%, and 46.9%, respectively, confirming the rationality and validity of our classification.

Finally, all the aforementioned results were integrated as a detailed LLNM risk stratification flow chart for PTC patients with CLNM. All patients exhibiting no fewer than five positive CLN, PTMC and LPTC patients who exhibited fewer than five positive CLN and with a total score no less than 100 and 150 based on their respective nomograms are considered as at a high-risk of LLNM. For those patients, a closer follow-up scheme should be conducted as the first choice. Prophylactic LLND could be considered as a second choice after synthesizing the patients’ preference as well as the surgeon’s overall assessment to address the relatively high LLNM risk. However, for those with low LLNM risk (PTMC and LPTC patients who exhibited fewer than five positive CLN and with total score of 0 and less than 100 based on their respective nomograms), neck dissection for the positive central region is enough, and no intervention of lateral neck region is needed. For patients in the moderate-risk group, the patients’ basic physical status and preference as well as the surgeon’s overall assessment should be comprehensively considered for individual treatment decisions.

## 5. Limitation

There are two main limitations of our study. Considering the retrospective nature, selection bias is inevitably produced. Furthermore, external validation should also be added to further validate our prediction model. Therefore, we aim to conduct more reliable multicentric, large sample, prospective, randomized control studies on this topic in the future.

## 6. Conclusions

A detailed stratification flow chart for PTC patients with CLNM to quantitatively assess LLNM risk was established, which may aid in clinical decision-making for those patients.

## Figures and Tables

**Figure 1 jcm-11-04975-f001:**
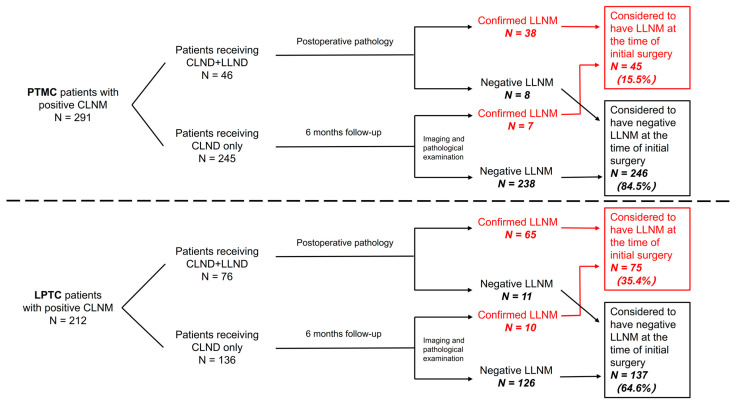
Flow diagram of case selection steps and follow-up information for patients enrolled. CLNM, central lymph node metastases; LLNM, lateral lymph node metastases; PTMC, papillary thyroid microcarcinoma; LPTC, large papillary thyroid carcinoma; CLND, central lymph node dissection; LLND, lateral lymph node dissection.

**Figure 2 jcm-11-04975-f002:**
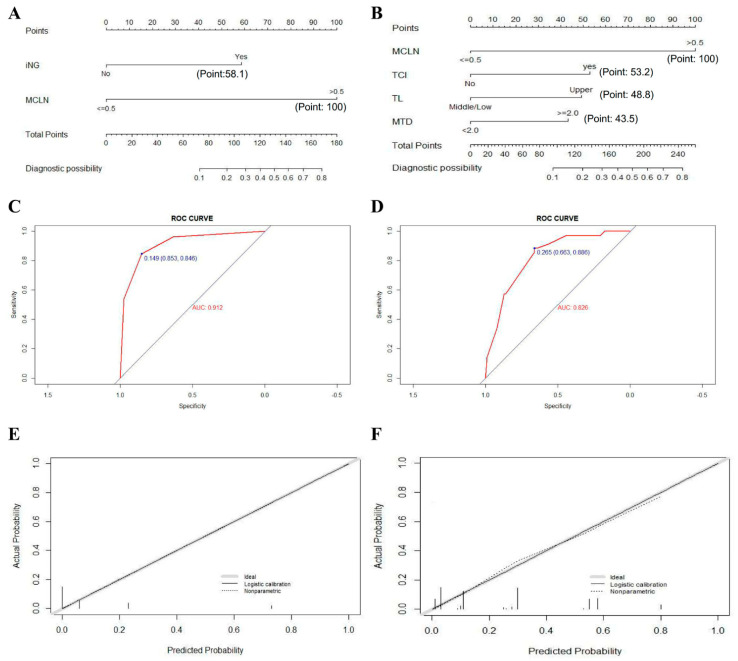
Construction, assessment, and validation of the predictive model. (**A**,**B**) The nomograms for predicting LLNM risk in PTMC and LPTC patients with fewer than five positive CLN, respectively; (**C**,**D**) The ROC curve and AUC of the nomograms for predicting LLNM risk in PTMC and LPTC patients with fewer than five positive CLN, respectively; (**E**,**F**) The calibration curves of the nomogram for predicting LLNM risk in PTMC and LPTC patients with fewer than five positive CLN, respectively. Actual probability is plotted on the y-axis, and the nomogram predicted probability on the x-axis. LLNM, lateral lymph node metastases; PTMC, papillary thyroid microcarcinoma; LPTC, large papillary thyroid carcinoma; CLN, central lymph nodes; iNG, ipsilateral nodular goiter; MCLN, maximum diameter of positive CLN; TCI, Thyroid capsular invasion; TL, tumor location; MTD, maximum tumor diameter; ROC, receiver operating characteristics.

**Figure 3 jcm-11-04975-f003:**
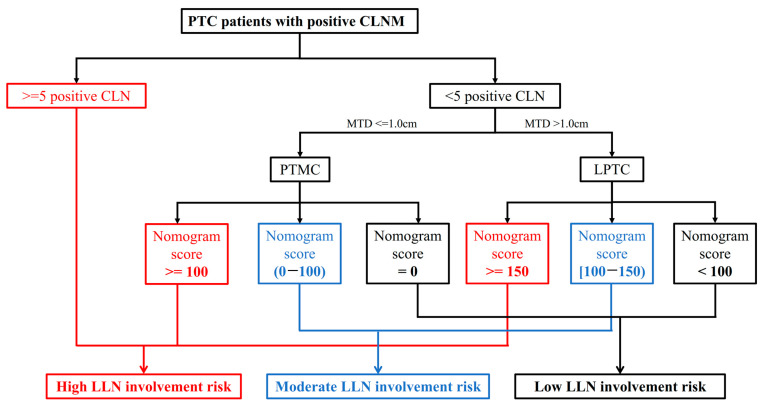
Detailed LLNM risk stratification flow chart for all PTC patients with positive CLNM. LLNM, lateral lymph node metastases; PTC, papillary thyroid carcinoma; CLNM, central lymph node metastases; PTMC, papillary thyroid microcarcinoma; LPTC, large papillary thyroid carcinoma; CLN, central lymph node; LLN, lateral lymph node.

**Table 1 jcm-11-04975-t001:** The clinicopathological characteristics of patients with different tumor size.

	All Patients		PTMC		LPTC		
	*n* = 503	%	*n* = 291	%	*n* = 212	%	*p* Value
Age (mean ± SD)	39.77 ± 11.83		40.32 ± 11.72		39.03 ± 11.97		0.228
BMI (mean ± SD)	23.91 ± 4.07		23.76 ± 4.09		24.12 ± 4.05		0.334
Maximum tumor diameter (mean ± SD)	1.08 ± 0.80		0.59 ± 0.19		1.74 ± 0.85		0.000
Gender							0.519
Male	217	43.1	122	41.9	95	44.8	
Female	286	56.9	169	58.1	117	55.2	
History of smoking							0.026
No	458	91.1	272	93.5	186	87.7	
Yes	45	8.9	19	6.5	26	12.3	
History of alcoholism							0.598
No	464	92.2	270	92.8	194	91.5	
Yes	39	7.8	21	7.2	18	8.5	
History of hypertension							0.660
No	418	83.1	240	82.5	178	84.0	
Yes	85	16.9	51	17.5	34	16.0	
History of diabetes							0.895
No	480	95.4	278	95.5	202	95.3	
Yes	23	4.6	13	4.5	10	4.7	
Thyroid capsular invasion							0.000
No	213	42.3	149	51.2	64	30.2	
Yes	290	57.7	142	48.8	148	69.8	
Bilateral disease							0.055
Absent	371	73.8	224	77.0	147	69.3	
Present	132	26.2	67	23.0	65	30.7	
Multifocality							0.880
Absent	278	55.3	160	55.0	118	55.7	
Present	225	44.7	131	45.0	94	44.3	
Tumor location							0.403
Upper portion	135	26.8	74	25.4	61	28.8	
Middle/Lower portion	368	73.2	217	74.6	151	71.2	
Number of positive CLN							0.000
1–2	256	50.9	171	58.8	85	40.1	
3–4	117	23.3	66	22.7	51	24.1	
≥5	130	25.8	54	18.6	76	35.8	
Maximum diameter of positive CLN							0.000
<1.0 cm	389	77.3	245	84.2	144	67.9	
≥1.0 cm	114	22.7	46	15.8	68	32.1	
PTC with ipsilateral Hashimoto thyroiditis							0.483
No	408	81.1	233	80.1	175	82.5	
Yes	95	18.9	58	19.9	37	17.5	
PTC with ipsilateral nodular goiter							0.527
No	354	70.4	208	71.5	146	68.9	
Yes	149	29.6	83	28.5	66	31.1	
LLN involvement							0.000
No	383	76.1	246	84.5	137	64.6	
Yes	120	23.9	45	15.5	75	35.4	

PTMC, papillary thyroid microcarcinoma; LPTC, large papillary thyroid carcinoma; CLN, central lymph nodes; LLN, lateral lymph nodes.

**Table 2 jcm-11-04975-t002:** Comparisons between patients with and without LLNM in PTMC and LPTC patient groups.

	All Patients (*n* (%))			PTMC (*n* (%))			LPTC (*n* (%))		
	Non-LLNM	LLNM		Non-LLNM	LLNM		Non-LLNM	LLNM	
	*n* = 383	*n* = 120	*p* Value	*n* = 246	*n* = 45	*p* Value	*n* = 137	*n* = 75	*p* Value
Age (mean ± SD)	40.41 ± 11.89	37.75 ± 11.47	0.032	40.63 ± 11.87	38.58 ± 10.83	0.280	40.00 ± 11.95	37.25 ± 11.87	0.110
BMI (mean ± SD)	23.96 ± 4.16	23.76 ± 3.82	0.644	23.78 ± 4.16	23.68 ± 3.74	0.888	24.29 ± 4.14	23.81 ± 3.89	0.413
Maximum tumor diameter (mean ± SD)	0.94 ± 0.62	1.50 ± 1.11	0.000	0.59 ± 0.19	0.62 ± 0.21	0.435	1.58 ± 0.62	2.03 ± 1.09	0.000
Gender			0.051			0.174			0.205
Male	156 (40.7)	61 (50.8)		99 (40.2)	22 (48.9)		57 (41.6)	38 (50.7)	
Female	227 (59.3)	59 (49.2)		147 (59.8)	23 (51.1)		80 (58.4)	37 (49.3)	
History of smoking			0.118			0.176			0.725
No	353 (92.2)	105 (87.5)		232 (94.3)	40 (88.9)		121 (88.3)	65 (86.7)	
Yes	30 (7.8)	15 (12.5)		14 (5.7)	5 (11.1)		16 (11.7)	10 (13.3)	
History of alcoholism			0.507			0.887			0.400
No	355 (92.7)	109 (90.8)		228 (92.7)	42 (93.3)		127 (92.7)	67 (89.3)	
Yes	28 (7.3)	11 (9.2)		18 (7.3)	3 (6.7)		10 (7.3)	8 (10.7)	
History of hypertension			0.042			0.421			0.049
No	311 (81.2)	107 (89.2)		201 (81.7)	39 (86.7)		110 (80.3)	68 (90.7)	
Yes	72 (18.8)	13 (10.8)		45 (18.3)	6 (13.3)		27 (19.7)	7 (9.3)	
History of diabetes			0.807			0.428			0.754
No	365 (95.3)	115 (95.8)		234 (95.1)	44 (97.8)		131 (95.6)	71 (94.7)	
Yes	18 (4.7)	5 (4.2)		12 (4.9)	1 (2.2)		6 (4.4)	4 (5.3)	
Thyroid capsular invasion			0.001			0.190			0.038
No	178 (46.5)	85 (70.8)		130 (52.8)	19 (42.2)		48 (35.0)	16 (21.3)	
Yes	205 (53.5)	35 (29.2)		116 (47.2)	26 (57.8)		89 (65.0)	59 (78.7)	
Bilateral disease			0.012			0.030			0.349
Absent	293 (76.5)	78 (65.0)		195 (79.3)	29 (64.4)		98 (71.5)	49 (65.3)	
Present	90 (23.5)	42 (35.0)		51 (20.7)	16 (35.6)		39 (28.5)	26 (34.7)	
Multifocality			0.017			0.000			0.717
Absent	223 (58.2)	55 (45.8)		148 (60.2)	12 (26.7)		75 (54.7)	43 (57.3)	
Present	160 (41.8)	65 (54.2)		98 (39.8)	33 (73.3)		62 (45.3)	32 (42.7)	
Tumor location			0.038			0.836			0.019
Upper portion	94 (24.5)	41 (34.2)		62 (25.2)	12 (26.7)		105 (76.6)	46 (61.3)	
Middle/Lower portion	289 (75.5)	79 (65.8)		184 (74.8)	33 (73.3)		32 (23.4)	29 (38.7)	
Number of positive CLN			0.000			0.000			0.000
1–2	220 (57.4)	36 (30.0)		156 (63.4)	15 (33.3)		64 (46.7)	21 (28.0)	
3–4	92 (24.0)	25 (20.8)		55 (22.4)	11 (24.4)		37 (27.0)	14 (18.7)	
≥5	71 (18.5)	59 (49.2)		35 (14.2)	19 (42.2)		36 (26.3)	40 (53.3)	
Maximum diameter of positive CLN			0.000			0.000			0.000
<1.0 cm	353 (92.2)	36 (30.0)		234 (95.1)	11 (24.4)		119 (86.9)	25 (33.3)	
≥1.0 cm	30 (7.8)	84 (70.0)		12 (4.9)	34 (75.6)		18 (13.1)	50 (66.7)	
PTC with ipsilateral Hashimoto thyroiditis			0.327			0.990			0.242
No	307 (80.2)	101 (84.2)		197 (80.1)	36 (80.0)		110 (80.3)	65 (86.7)	
Yes	76 (19.8)	19 (15.8)		49 (19.9)	9 (20.0)		27 (19.7)	10 (13.3)	
PTC with ipsilateral nodular goiter			0.000			0.000			0.003
No	293 (76.5)	61 (50.8)		189 (76.8)	19 (42.2)		104 (75.9)	42 (56.0)	
Yes	90 (23.5)	59 (49.2)		57 (23.2)	26 (57.8)		33 (24.1)	33 (44.0)	

PTMC, papillary thyroid microcarcinoma; LPTC, large papillary thyroid carcinoma; CLN, central lymph nodes; PTC, papillary thyroid carcinoma.

**Table 3 jcm-11-04975-t003:** Surgery-related complications after CLND alone and CLND + LLND.

Surgical Complications	CLND Alone	%	CLND + LLND	%	*p*-Value
Patients	381		122		
Postoperative hypoparathyroid hormone	90		35		
Temporary	83	21.8	29	23.8	0.646
Permanent	7	1.8	6	4.9	0.062
Postoperative hoarseness	42		21		
Temporary	36	9.4	16	13.1	0.247
Permanent	6	1.6	5	4.1	0.097
Chyle leakage	1	0.2	4	3.3	0.003
Relatively severe surgery-related complications					
(Permanent hypoparathyroid hormone + Permanent hoarseness + Chyle leakage)					
	14	3.7	15	12.3	0.000

CLND, central lymph node dissection; LLND, lateral lymph node dissection; Postoperative hypoparathyroid hormone: blood parathyroid hormone (PTH) lower than 15.0 pg/mL; Temporary hypoparathyroid hormone: having hypoparathyroid hormone on postoperative day 1, yet returning to normal within 1 month after surgery; Temporary hoarseness: having hoarseness on postoperative day 1, yet returning to normal within 6 months after surgery.

**Table 4 jcm-11-04975-t004:** Univariate and multivariate analyses for PTMC patients with fewer than five positive CLN.

	Univariate Analysis		Multivariate Analysis	
	Hazard Ratio (95% CI)	*p* Value	Hazard Ratio (95% CI)	*p* Value
Factors selected				
Multifocality		0.007		0.089
Yes vs. No	3.221 (1.370–7.573)		2.584 (0.865–7.724)	
PTC with ipsilateral nodular goiter		0.000		0.000
Yes vs. No	5.926 (2.490–14.106)		8.678 (2.758–27.312)	
Number of positive CLN		0.043		0.563
(2,4) vs. 1	2.678 (1.034–6.938)		1.434 (0.423–4.859)	
Maximum diameter of positive CLN		0.000		0.000
>0.5 cm vs. ≤0.5 cm	31.935 (10.301–99.007)		36.046 (10.039–129.423)	
Age		0.546		
≥55 vs. <55	0.679 (0.193–2.389)			
BMI		1.000		
>23 vs. ≤23	1.000 (0.443–2.259)			
Maximum tumor diameter		0.577		
≥0.5 cm vs. <0.5 cm	1.339 (0.480–3.731)			
Gender		0.699		
Male vs. Female	1.177 (0.515–2.688)			
Thyroid capsular invasion		0.262		
Yes vs. No	1.603 (0.703–3.653)			
Tumor location		0.464		
Upper vs. Middle/Lower	1.394 (0.572–3.397)			
PTC with ipsilateral Hashimoto thyroiditis		0.741		
Yes vs. No	0.828 (0.270–2.541)			
Bilateral disease		0.091		
Yes vs. No	2.130 (0.887–5.114)			

PTMC, papillary thyroid microcarcinoma; PTC, papillary thyroid carcinoma; CLN, central lymph nodes; BMI, Body Mass Index.

**Table 5 jcm-11-04975-t005:** Risk stratification of PTMC and LPTC patients with fewer than five positive CLNM.

		Low Risk (TS = 0)	Moderate Risk (0 < TS < 100)	Relatively High Risk (TS ≥ 100)	
		(*n* = 135, %)	(*n* = 49, %)	(*n* = 53, %)	*p* Value
PTMC	Negative LLNM	134 (99.3)	46 (93.9)	31 (58.5)	0.000
	Positive LLNM	1 (0.7)	3 (6.1)	22 (41.5)	
		**Low Risk (TS < 100)**	**Moderate Risk (100 ≤ TS < 150)**	**Relatively High Risk (TS ≥ 150)**	
		**(*n* = 46, %)**	**(*n* = 26, %)**	**(*n* = 64, %)**	***p* Value**
LPTC	Negative LLNM	45 (97.8)	22 (84.6)	34 (53.1)	0.000
	Positive LLNM	1 (2.2)	4 (15.4)	30 (46.9)	

PTMC, papillary thyroid microcarcinoma; LPTC, large papillary thyroid carcinoma; CLNM, central lymph node metastases, LLNM, lateral lymph node metastases; TS, total score.

**Table 6 jcm-11-04975-t006:** Univariate and multivariate analyses for LPTC patients with fewer than five CLN.

	Univariate Analysis		Multivariate Analysis	
	Hazard Ratio (95% CI)	*p* Value	Hazard Ratio (95% CI)	*p* Value
Factors selected				
Tumor location		0.011		0.017
Upper vs. Middle/Lower	2.856 (1.269–6.429)		3.244 (1.236–8.517)	
Thyroid capsular invasion		0.005		0.041
Yes vs. No	4.875 (1.598–14.877)		3.553 (1.056–11.955)	
Maximum tumor diameter		0.043		0.042
≥2.0 cm vs. <2.0 cm	2.877 (1.242–6.668)		2.819 (1.039–7.646)	
Maximum diameter of positive CLN		0.000		0.000
>0.5 cm vs. ≤0.5 cm	11.778 (3.387–40.953)		10.924 (2.985–39.978)	
Age		0.637		
≥55 vs. <55	1.286 (0.452–3.654)			
BMI		0.286		
>23 vs. ≤23	1.530 (0.701–3.340)			
Number of positive CLN		0.332		
(2,4) vs. 1	1.532 (0.647–3.627)			
Gender		0.257		
Male vs. Female	1.566 (0.721–3.400)			
PTC with ipsilateral nodular goiter		0.060		
Yes vs. No	2.163 (0.968–4.836)			
Multifocality		0.597		
Yes vs. No	1.232 (0.568–2.674)			
PTC with ipsilateral Hashimoto thyroiditis		0.470		
Yes vs. No	0.675 (0.233–1.960)			
Bilateral disease		0.572		
Yes vs. No	1.283 (0.540–3.047)			

LPTC, large papillary thyroid carcinoma; PTC, papillary thyroid carcinoma; CLN, central lymph nodes; BMI, Body Mass Index.

## Data Availability

The datasets generated and analyzed during the current study are available from the corresponding author on reasonable request.

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
