# Peer review of "Lateral Involvement in Different Sized Papillary Thyroid Carcinomas Patients with Central Lymph Node Metastasis: A Multi-Center Analysis"

_jcm, 2022, doi:10.3390/jcm11174975_

Round 1

Reviewer 1 Report

This is a retrospective study that analyzes the probability of lateral neck lymph node metastases in papillary thyroid cancer. The authors specifically focus on patients with less than 5 positive central neck lymph node metastases. They identify the risk factors of lateral neck lymph node metastasis in these patients and establish a risk prediction model (nomogram) to stratify patients into low, moderate, and high risk for lateral neck lymph node metastases. This is an interesting study with practical implication. Please address the following concerns.

1. The authors identify risk factors of lateral neck lymph node metastasis in patients with less than 5 positive central neck lymph nodes and use these factors to construct the prediction nomogram model. Can author provide the exact points for each risk factor? For example, the point of ipsilateral nodular goiter(iNG) in PTMC patients is somewhere around 50-60 (Figure 2A). In this way, the readers can easily apply this model, calculate the total risk score, and allocate their own patients into low, moderate, and high risk.

2. In the study, PTC with ipsilateral nodular goiter was an independent risk factors for lateral neck lymph node metastasis in PTMC patients with <5 positive central neck lymph node metastasis (Table 4). Is there any postulation or hypothesis to explain why ipsilateral nodular goiter was associated with lateral neck lymph node metastasis?

3. In this study, 280 patients received thyroid lobectomy and 223 patients received total thyroidectomy. Among patients who underwent total thyroidectomy, did the central neck dissection performed ipsilateral or bilaterally? If bilateral central neck dissections were performed in patients who underwent total thyroidectomy, it would theoretically harvest more central lymph nodes than patients underwent lobectomy and unilateral central neck dissection. Total thyroidectomy patients might tend to have positive central lymph node >=5, which is a risk factor in this study. Can you tell us the operation type (lobectomy vs total thyroidectomy) in patients have positive central lymph node metastases >=5? How do you adjust this bias (discrepancy regarding the number of lymph nodes harvested between unilateral and bilateral central neck dissection)?

4. Table 2 mainly demonstrates the comparison between patients with and without lateral neck lymph node metastases. The title of the table 2 should represent this content more precisely. I think the current title (Table 2. The clinicopathological characteristics of patients with different tumor size.) might be more suitable for Table 1, please consider.

5. In Figure 1, the left side showed two groups of patients: PTMC patients with “positive” CLNM and LPTC patients with CLNM. The author may consider adding “positive” in the second group for consistency or deleting the word” positive” in the first group.

Author Response

Dear Editors and Reviewers:

On behalf of my co-authors, I would like to express appreciation for giving us an opportunity to revise our manuscript, “Lateral Involvement in Different Sized Papillary Thyroid Carcinomas Patients with Central Lymph Node Metastasis: a Multi-center Analysis” (Manuscript ID: jcm-1853525).

We have studied these comments carefully and tried our best to revise and improve the manuscript.

Revised portions are marked red in the paper. The main corrections and our response to the reviewer’s comments are as following:

Reviewer #1 comments:

  1. Response to comment: "The authors identify risk factors of lateral neck lymph node metastasis in patients with less than 5 positive central neck lymph nodes and use these factors to construct the prediction nomogram model. Can author provide the exact points for each risk factor? For example, the point of ipsilateral nodular goiter(iNG) in PTMC patients is somewhere around 50-60 (Figure 2A). In this way, the readers can easily apply this model, calculate the total risk score, and allocate their own patients into low, moderate, and high risk. "  

Response: We appreciate the Reviewer for helping us capture this element that we ignored before. We have marked the risk point for each risk factor, and added this information in Figure 2A as the Reviewer suggested.

  1. Response to comment: "In the study, PTC with ipsilateral nodular goiter was an independent risk factors for lateral neck lymph node metastasis in PTMC patients with <5 positive central neck lymph node metastasis (Table 4). Is there any postulation or hypothesis to explain why ipsilateral nodular goiter was associated with lateral neck lymph node metastasis? "  

Response: It is a great honor to answer this question. To our knowledge, few previous studies have revealed the association between ipsilateral nodular goiter and lateral neck lymph node metastasis in patients with PTMC, and no explanations for this phenomenon exist. We think this implies that the coexistence of nodular goiter portends a more aggressive tumor. Mention of this phenomenon by the reviewer has been valuable to us, and we will be investigating their underlying mechanism in terms of PTMC metastasis in the future.

  1. Response to comment: "In this study, 280 patients received thyroid lobectomy and 223 patients received total thyroidectomy. Among patients who underwent total thyroidectomy, did the central neck dissection performed ipsilateral or bilaterally? If bilateral central neck dissections were performed in patients who underwent total thyroidectomy, it would theoretically harvest more central lymph nodes than patients underwent lobectomy and unilateral central neck dissection. Total thyroidectomy patients might tend to have positive central lymph node >=5, which is a risk factor in this study. Can you tell us the operation type (lobectomy vs total thyroidectomy) in patients have positive central lymph node metastases >=5? How do you adjust this bias (discrepancy regarding the number of lymph nodes harvested between unilateral and bilateral central neck dissection)? "  

Response: The bilateral central neck dissections were performed in patients who underwent total thyroidectomy. Moreover, for patients with only unilateral tumor and received thyroid lobectomy, prophylactic bilateral central neck dissection was conducted for those with clinically detected contralateral central lymph nodes that were highly suspected as having tumor involvement using preoperative ultrasonography. Given this, we think the extent of central lymph node dissection is safe enough, and the number of positive centra lymph nodes metastases is independent of the number of lymph nodes harvested.   

  1. Response to comment: "Table 2 mainly demonstrates the comparison between patients with and without lateral neck lymph node metastases. The title of the table 2 should represent this content more precisely. I think the current title (Table 2. The clinicopathological characteristics of patients with different tumor size.) might be more suitable for Table 1, please consider. "  

Response: Thank you for your valuable suggestion. The title for Table 1 and Table 2 has been modified in our revised manuscript according to your advice.

Table 1. The clinicopathological characteristics of patients with different tumor size

Table 2. Comparisons between patients with and without LLNM for PTMC and LPTC patients

  1. Response to comment: "In Figure 1, the left side showed two groups of patients: PTMC patients with “positive” CLNM and LPTC patients with CLNM. The author may consider adding “positive” in the second group for consistency or deleting the word “positive” in the first group. "  

Response: We appreciate the Reviewer for helping us capture this element that we ignored before. We have modified Figure 1 in our revised manuscript as per the Reviewer’s suggestion.

At last, we appreciate for Editors/Reviewers’ input earnestly, and hope that the correction will meet with approval.

Looking forward to hearing from you.

Yours sincerely,

Lei Tao & Yu Heng

Reviewer 2 Report

I have read with great interest the manuscript entitled “Lateral Involvement in Different Sized Papillary Thyroid Carcinomas Patients with Central Lymph Node Metastasis: a Multi-center Analysis”. In this study the authors used a cohort of 503 patients that underwent thyroidectomy and CLND and aimed to identify the risk stratification of lateral LNM. Of the 291 patients with PTMC the risk for LLNM was 15.5% and of the 212 patients with large PTC the risk was 35.4%. Patients with more than 5 CLNM were placed at high-risk for LLNM. A nomogram was created for patients with less than 5 CLNM and was based on tumor location, diameter, LNM diameter, capsular invasion. This is a well-designed study that presents lengthy but important results.

Comments –

1.     Abstract – start your results section with the risk for LLNM in the PTMC and LPTC groups.

2.     Terminology – the more common term is extrathyroidal extension (ETE) rather the thyroid capsular invasion.

3.     Must define all abbreviations at all figures. Especially in figure 2 – iNG, MCLN, TCI, TL, MTD.

4.     Thank you for adding the detailed complications table. Authors should discuss the risk-benefit issue of prophylactic LLND in face of increased complications. It is unclear that all occult LLNM will grow over time and require future surgery. Also, why not leave the lateral compartment for future surgery if it will present in the future as that lateral compartment will be a virgin compartment and thus dose not pose the risks of redo surgery in the neck?

5.     I completely disagree with the sentence on row 348 “in view of the extremely disease specific mortality” – no such data were given in the referenced article and to the best of my knowledge there is no higher mortality in delayed surgery (as opposed to prophylactic) of the lateral neck. This sentence must be omitted and the entire paragraph should be revised to reflect comment number 4 as one must weigh the risks of surgery to the benefit of prophylactic LLND.

6.     A limitation paragraph must be added.

Author Response

Dear Editors and Reviewers:

On behalf of my co-authors, I would like to express appreciation for giving us an opportunity to revise our manuscript, “Lateral Involvement in Different Sized Papillary Thyroid Carcinomas Patients with Central Lymph Node Metastasis: a Multi-center Analysis” (Manuscript ID: jcm-1853525).

We have studied these comments carefully and tried our best to revise and improve the manuscript.

Revised portions are marked red in the paper. The main corrections and our response to the reviewer’s comments are as following:

Reviewer #2 comments:

  1. Response to comment: "Abstract – start your results section with the risk for LLNM in the PTMC and LPTC groups. "  

Response: Thank you for your valuable suggestion. The related information has been added in “Abstract” section in our revised manuscript as per your advice.

  1. Response to comment: "Terminology – the more common term is extrathyroidal extension (ETE) rather the thyroid capsular invasion. "  

Response: Thank you for bringing ETE to our attention. In regards to this terminology, we have found several articles using the phrase ‘thyroid capsular invasion’, and our members are more accustomed to this phrase as well. Thus, if possible, may we request that this terminology to be left unchanged? Much thanks!

  1. Response to comment: "Must define all abbreviations at all figures. Especially in figure 2 – iNG, MCLN, TCI, TL, MTD. "  

Response: We appreciate the Reviewer for helping us capture this element that we ignored before. The Figure legend has been refined in our revised manuscript.

  1. Response to comment: "Thank you for adding the detailed complications table. Authors should discuss the risk-benefit issue of prophylactic LLND in face of increased complications. It is unclear that all occult LLNM will grow over time and require future surgery. Also, why not leave the lateral compartment for future surgery if it will present in the future as that lateral compartment will be a virgin compartment and thus dose not pose the risks of redo surgery in the neck? "  

Response: It is a great honor to answer this question. Our clinical prediction system provides an early warning for occult LLNM. Prophylactic neck dissection is only one means of active intervention; other options include prophylactic RAI treatment and closer follow-up. For high-risk patients defined by our model, it can play an early warning role, and specific medical decisions still need to be made by clinicians.

  1. Response to comment: "I completely disagree with the sentence on row 348 “in view of the extremely disease specific mortality” – no such data were given in the referenced article and to the best of my knowledge there is no higher mortality in delayed surgery (as opposed to prophylactic) of the lateral neck. This sentence must be omitted and the entire paragraph should be revised to reflect comment number 4 as one must weigh the risks of surgery to the benefit of prophylactic LLND. "  

Response: We appreciate the Reviewer’s valuable advice. Our description of “extremely disease specific mortality” is not appropriate as the reviewer has pointed out, and the related content has been omitted in our revised manuscript. For patients defined as having high LLNM risk, a closer follow-up scheme should be conducted as the first choice. Prophylactic LLND could be considered as a second choice after synthesizing the patients’ preference as well as the surgeon’s overall assessment to address the relatively high LLNM risk.  

  1. Response to comment: "A limitation paragraph must be added. "  

Response: We are very sorry for our negligence. We have supplemented a “Limitation” section after “Conclusion” in our revised manuscript as per the Reviewer’s suggestion.

At last, we appreciate for Editors/Reviewers’ warm work earnestly, and hope that the correction will meet with approval.

Looking forward to hearing from you.

Yours sincerely,

Lei Tao & Yu Heng

Round 2

Reviewer 2 Report

Comments were addressed. 

Limitations paragraph should be edited for style and English grammar. Also, this paragraph belongs before the conclusions. 

Author Response

Dear Editors and Reviewers:

On behalf of my co-authors, I would like to express appreciation for giving us an opportunity to revise our manuscript, “Lateral Involvement in Different Sized Papillary Thyroid Carcinomas Patients with Central Lymph Node Metastasis: a Multi-center Analysis” (Manuscript ID: jcm-1853525).

We have studied these comments carefully and tried our best to revise and improve the manuscript.

Revised portions are marked red in the paper. The main corrections and our response to the reviewer’s comments are as following:

Reviewer #1 comments:

  1. Response to comment: "Limitations paragraph should be edited for style and English grammar. Also, this paragraph belongs before the conclusions. "  

Response: We are very sorry for our numerous awkward English expressions of “Limitations” paragraph. We have consulted an English polishing service. We sincerely hope our efforts in improving the English will meet your approval.

Also, we have brought this paragraph forward in our revised manuscript.

At last, we appreciate for Editors/Reviewers’ input earnestly, and hope that the correction will meet with approval.

Looking forward to hearing from you.

Yours sincerely,

Lei Tao & Yu Heng
